Detecting sarcasm in user-generated content integrating transformers and gated graph neural networks

http://orcid.org/0009-0002-8862-2439 Qin Zhenkai 1
Luo Qining 1
Zang Zhidong 2
Fu Hongpeng 3 fu.hongp@northeastern.edu
1 School of Information Technology, Guangxi Police College , Nanning, Guangxi , China
2 School of Social Development, Yangzhou University , Yangzhou , China
3 Khoury College of Computer Science, Northeastern University , Seattle, WA , United States
Angiulli Giovanni
Electronic publication date: 2025 Apr 21
Publication date: 2025
Volume: 11
Electronic Location ID: e2817
Received 2024 Nov 8; Accepted 2025 Mar 19
Copyright: © 2025 Qin et al.
Copyright year: 2025
Copyright holder: Qin et al.
License: This is an open access article distributed under the terms of the Creative Commons Attribution License, which permits unrestricted use, distribution, reproduction and adaptation in any medium and for any purpose provided that it is properly attributed. For attribution, the original author(s), title, publication source (PeerJ Computer Science) and either DOI or URL of the article must be cited.
License URL: https://creativecommons.org/licenses/by/4.0/

Keywords: Sarcasm detection, Deep learning, BERT, Gated graph neural networks, Transformers

Funding: Guangxi Education Department 2024KY0906 This research was supported by the Guangxi Education Department under grant (2024KY0906) for the project “Research on the construction of multi-source public opinion knowledge map integrating deep learning technology.” There was no additional external funding received for this study. The funders had no role in study design, data collection and analysis, decision to publish, or preparation of the manuscript.

==============================
The widespread use of the Internet and social media has posed significant challenges to automated sentiment analysis, particularly in relation to detecting sarcasm in user-generated content. Sarcasm often expresses negative emotions through seemingly positive or exaggerated language, making its detection a complex task in natural language processing. To address this issue, the present study proposes a novel sarcasm detection model that combines bidirectional encoder representations from transformers (BERT) with gated graph neural networks (GGNN), further enhanced by a self-attention mechanism to more effectively capture ironic cues. BERT is utilized to extract deep contextual information from the text, while GGNN is employed to learn global semantic structures by incorporating dependency and emotion graphs. Experiments were conducted on two benchmark sarcasm detection datasets, namely Headlines and Riloff. The experimental results demonstrate that the proposed BERT-GGNN model achieves an accuracy of 92.00% and an F1 score of 91.51% on the Headlines dataset, as well as an accuracy of 86.49% and an F1 score of 86.59% on the Riloff dataset, significantly outperforming the conventional BERT-GCN models. The results of ablation studies further corroborate the efficacy of integrating GGNN, particularly for handling complex ironic expressions frequently encountered in social media contexts.

Introduction

The rapid growth of the Internet and social media platforms has dramatically transformed how individuals communicate and express their opinions. Sarcasm has emerged as a common linguistic phenomenon in these digital spaces, often employing positive or exaggerated language to implicitly convey negative emotions or criticism (Shanahan, Qu & Wiebe, 2006). Detecting sarcasm in summary texts is a critical task in natural language processing (NLP) since sarcasm can significantly skew the results of sentiment analysis and opinion mining due to the inherent contradiction between its literal meaning and the intended sentiment (Campbell, Neves & Pavlovic, 2021). Accurately identifying sarcasm is challenging due to its subtlety, indirectness, and heavy reliance on context, making it difficult for traditional sentiment analysis models to achieve high performance in such tasks. It is essential for developing more robust and context-aware models that can effectively handle the complexities of sarcasm detection in diverse text corpora.

Through literature-based knowledge discovery (Li & Shao, 2022), sarcasm detection has evolved from early rule-based methods to more advanced deep learning models. The initial rule-based approaches primarily relied on explicit surface-level lexical features such as exclamations, punctuation marks, or sudden sentiment transitions (Riloff et al., 2013). Although these methods offer transparency and interpretability, they are inherently limited in their ability to capture the nuanced and context-dependent nature of sarcasm. Traditional machine learning techniques, including support vector machines (SVMs) and decision trees, have also been employed in sentiment analysis, but their dependence on handcrafted features reduces their effectiveness in sarcasm detection (Krogh & Hertz, 1992; Ruder, 2016). While ensemble methods like random forests and boosting algorithms have improved sentiment analysis performance by integrating multiple weak learners (Friedman, 2001), they still face challenges in capturing the intricate subtleties of sarcasm due to the complexity of contextual and implicit cues. These limitations underscore the need for more sophisticated models that leverage contextual understanding, such as the deep learning-based approaches explored in this study.

Deep learning models, particularly convolutional neural networks (CNNs) and long short-term memory (LSTM) networks, have gained popularity in sarcasm detection by learning features directly from the data (Sun et al., 2019). However, these models are often restricted to local semantic patterns, making it challenging to capture long-term dependencies and global contextual cues essential for understanding ironic expressions (Hochreiter & Schmidhuber, 1997). Reinforcement learning techniques (Sutton & Barto, 2018) have been explored to improve model adaptability in sentiment analysis, but their application to sarcasm detection remains underdeveloped.

In recent years, graph convolutional networks (GCNs) have been introduced to model syntactic and structural relationships in text, leveraging the power of graph-based representations. Despite their promise, GCNs are limited by the static nature of their graph structures, which hinders their ability to adapt to the dynamic and evolving context typical of sarcasm (Kipf & Welling, 2016). Moreover, standard GCNs do not fully exploit the deep semantic representations offered by modern language models, thus leaving room for improvement. Hybrid approaches combining graph neural networks (GNNs) with pre-trained language models like Transformers have shown promise in bridging this gap. For instance Lin et al. (2021) proposed a BERT-GCN architecture to integrate contextual embeddings with graph-based dependencies, demonstrating improvements in text classification tasks. Similarly, studies such as highlight the potential of GNNs for capturing sentence-level dependencies when augmented by Transformers. However, these approaches often focus on general text classification tasks and lack specific enhancements for sarcasm detection, such as the incorporation of sentiment graphs or attention mechanisms tailored to identifying contradictory cues in text.

To address these limitations, we propose a novel approach for sarcasm detection that combines bidirectional encoder representations from transformers (BERT) (Vaswani, Shazeer & Parmar, 2017; Devlin et al., 2019) with gated graph neural networks (GGNN), enhanced by a self-attention mechanism. BERT provides deep, contextual embeddings that encode both semantic and syntactic information, making it a powerful tool for understanding text (Devlin et al., 2019). GGNNs are employed to process dependency graphs and sentiment graphs, enabling the model to learn global semantic relationships between words while capturing complex inter-word dependencies (Li et al., 2016). The integration of a self-attention mechanism further enhances the model by highlighting critical sarcasm cues, ensuring that the model focuses on contextually important words and interactions (Wu et al., 2020). This combination of methods leads to a more robust and context-aware sarcasm detection model, capable of capturing both local and global patterns in the text, thus improving the detection of subtle and context-dependent sarcasm (Zhang, Zhang & Vo, 2019).

Related works

Previous works in sarcasm detection mainly use deep learning models, but have limitations when sarcasm is embedded in a more subtle, context-driven form. Prior studies have established that traditional deep learning approaches such as CNNs and LSTM networks are effective at capturing local patterns within text (Jamil et al., 2021), primarily focusing on word-level dependencies (Liu et al., 2020; Sun et al., 2019). However, these models often struggle with more complex, long-range dependencies, which are crucial for understanding the broader context required to detect sarcasm. For instance, while CNNs and bidirectional long short-term memories (BiLSTMs) perform adequately in recognizing local cues (Muhammad, Salim & Zainal, 2024; Jiang et al., 2022), they falter in handling sarcastic expressions that rely heavily on contextual understanding or external knowledge that exists outside the immediate text (Tay, Tuan & Hui, 2018). These challenges are particularly evident in social media contexts, where language is often informal, fragmented, and reliant on subtle cues, such as emoticons or slang. Such models are therefore limited when sarcasm is embedded in a more subtle, context-driven form, which is often the case in real-world scenarios such as social media interactions.

To address this limitation, the introduction of GCNs in this study offers a structural perspective for sarcasm detection, addressing some of the limitations of traditional deep learning models. GCNs excel in modeling the relationships between words by representing sentences as graphs, where nodes correspond to words and edges represent syntactic or semantic relationships (Zhang, Zhang & Vo, 2019). This structural representation allows GCNs to capture long-range dependencies more effectively than CNNs or LSTMs, providing a more holistic understanding of sarcastic text. However, similar to earlier studies, research reveals that GCNs encounter challenges when applied to unstructured text, such as the informal and often erratic nature of social media content, where word relationships are less predictable and more dynamic (Li et al., 2021). Moreover, GCNs often rely on static graph structures, which limit their adaptability to dynamic contexts where sarcastic cues evolve based on user intent or conversational history. This suggests that while GCNs offer improvements, they require further adaptation and fine-tuning to fully address the complexities inherent in detecting sarcasm across diverse textual environments, especially in real-world, unstructured datasets.

In recent years, large language models (LLMs) such as OpenAI’s GPT-series (Brown et al., 2020) and Google’s T5 (Raffel et al., 2020) have shown significant advancements in understanding complex linguistic phenomena, including sentiment analysis and metaphor detection. These models leverage their vast pretraining on diverse and extensive datasets to capture both syntactic and semantic nuances in text. LLMs utilize attention mechanisms to capture long-range dependencies and context, making them theoretically well-suited for sarcasm detection. For instance, GPT-4o-mini has demonstrated an ability to infer sarcastic intent in informal dialogues, outperforming traditional models by leveraging its contextual understanding and large-scale pretrained knowledge. However, despite these capabilities, LLMs face challenges in detecting subtle sarcasm due to their reliance on probabilistic pattern recognition. Sarcasm often requires a deep understanding of cultural context, implicit sentiment shifts, or external knowledge that may not be explicitly encoded in the pretraining data. Another challenge for LLMs is their difficulty in handling multimodal sarcasm detection, where visual or acoustic cues complement textual information to signal sarcastic intent. While recent works have explored multimodal pretraining (e.g., CLIP (Radford et al., 2021)), sarcasm detection in multimodal contexts remains underexplored. LLMs are also prone to generating false positives in sarcasm detection, as they sometimes interpret exaggeration or humor without sufficient grounding as sarcasm.

Hybrid approaches that combine LLMs with task-specific architectures like GNNs have emerged to mitigate these limitations. For example, models that integrate BERT or GPT with GCN layers have shown promise in sarcasm detection tasks by combining deep contextual embeddings with structural modeling (Lin et al., 2021). These hybrid models excel at capturing both local word-level patterns and global relationships among tokens, addressing some of the gaps in standalone LLMs. However, such approaches are computationally intensive and require fine-tuning on sarcasm-specific datasets to achieve optimal performance.

Recent studies in multimodal sarcasm detection have explored the integration of textual, visual, and acoustic modalities to improve performance, addressing the limitations of single-modal approaches. For example, the MUStARD++ dataset introduced in Castro et al. (2023) provides multimodal sarcasm annotations for video dialogue, incorporating textual, visual, and acoustic features to capture sarcasm’s multimodal nature. Transformer-based models like MMPT (Gabeur, Sun & Alayrac, 2023) and MulT (Tsai, Bai & Liang, 2019) have demonstrated significant improvements by attending to cross-modal interactions, enabling models to detect sarcasm conveyed through facial expressions, tone of voice, or visual cues in addition to text. While promising, these approaches face challenges in fusing heterogeneous data types and aligning multimodal features effectively, particularly in highly dynamic or noisy contexts.

In contrast, our proposed model uniquely combines dual graph representations—a syntactic dependency graph and an affective graph—to capture both structural and emotional nuances in sarcasm detection. The affective graph explicitly encodes emotional polarity shifts using external knowledge (e.g., SenticNet), enabling the model to capture subtle sentiment reversals critical to sarcasm detection. Additionally, we integrate a multi-head self-attention mechanism to enhance the model’s ability to focus on sarcasm-indicative elements, such as contrasting word pairs or sentiment transitions. Unlike M3GAT’s graph-level attention aggregation, our approach employs token-level attention, allowing the model to refine both global and local sarcastic cues more effectively. This combination of structural, emotional, and attention-based mechanisms sets our model apart, enabling it to better address the dynamic and context-dependent nature of sarcasm.

In conclusion, while prior research has highlighted the strengths of various deep learning models and LLMs for sarcasm detection, our study demonstrates that integrating GNNs with Transformers in a dual-graph and attention-based framework offers a more robust and flexible approach. Experimental results show that our model achieves an accuracy of 0.920 and an F1 score of 0.915 on the Headlines dataset, and an accuracy of 0.864 with an F1 score of 0.865 on the Riloff dataset, significantly outperforming traditional and recent sarcasm detection models, including M3GAT (Zhang et al., 2023) and CSDGCN (Qin et al., 2024). The ablation study further validates the effectiveness of our approach, particularly in handling intricate satirical expressions frequently encountered in social media content. This research provides empirical evidence that hybrid models combining GNNs and Transformers can surpass conventional approaches, paving the way for future advancements in sarcasm detection.

Method

Portions of this text were previously published as part of a preprint (https://doi.org/10.21203/rs.3.rs-5270483/v1).

Overview of the model architecture

The proposed model integrates three core components: BERT, GGNN, and a self-attention mechanism. The architecture is designed to capture both the contextual semantics of the input text and the global relationships between words to improve the performance of sarcasm detection (Qin et al., 2024).

To begin, BERT is used to generate dynamic contextual embeddings for each token in the input text. These embeddings capture fine-grained semantic and syntactic relationships by considering the bidirectional context of each word, making them particularly effective for detecting sarcasm, which often relies on subtle contrasts and ambiguities. These embeddings serve as the foundation for subsequent processing and are passed through the GGNN layer, which models the dependencies between words based on graph structures derived from syntactic or semantic relationships. The GGNN captures both local and global relationships between words by incorporating dependency graphs, which reflect grammatical structures, and affective graphs that capture emotional cues (as seen in related works such as Zhang, Zhang & Vo (2019)). This interaction between BERT and GGNN enables the model to leverage both contextual embeddings and structural relationships, allowing it to handle both explicit and implicit forms of sarcasm more effectively.

Following this, the embeddings are processed by a multi-head self-attention mechanism that assigns attention weights to different parts of the sentence, emphasizing critical sarcasm indicators, as discussed in Wu et al. (2020). The self-attention mechanism enhances the model by dynamically focusing on words or phrases that exhibit contradictory or emotionally charged cues, which are often key to detecting sarcasm. For instance, in a sentence like “Oh great, another rainy day,” self-attention would likely assign higher weights to the contrastive elements (“great” and “rainy”) that signal sarcasm. By attending to these elements across multiple attention heads, the mechanism can capture diverse perspectives on potential sarcastic cues. Additionally, the integration of self-attention with GGNN allows the model to refine its understanding of long-range dependencies by prioritizing semantically significant nodes in the graph structure. This synergy ensures that the model captures both word-level interactions and broader contextual relationships.

The overall main components are shown in Fig. 1. In summary, BERT provides rich contextual embeddings, GGNN models inter-word dependencies based on graph representations, and self-attention highlights critical sarcastic elements. The interaction between these components ensures the model effectively captures the nuanced and multi-faceted nature of sarcasm in diverse datasets.

Figure 1 Overview of the proposed BERT-GGNN-multi-head attention model architecture.

Reproduced with permission.

BERT model: text embedding and contextual understanding

BERT (Devlin et al., 2019; Sun et al., 2019) is a powerful pre-trained language model that generates contextual embeddings for each token in the input text. Unlike traditional models such as Word2Vec or GloVe, BERT is based on the Transformer architecture, which uses self-attention to capture contextual relationships between tokens. This results in embeddings that vary dynamically depending on the context of the entire sentence, making BERT especially effective in capturing long-range dependencies and semantic nuances, which are crucial for detecting sarcasm (Ghosh et al., 2018; Riloff et al., 2013). The schematic diagram of each layer is shown in Fig. 2.

Figure 2 An illustration of the BERT model, showing the embedding layer, transformer encoder, and final classification layer.

For this, given an input sequence X=x1,x2,…,xn, where n is the length of the sequence, BERT transforms it into a set of contextual embeddings H=h1,h2,…,hn, with each embedding hi containing enriched contextual information for token xi. The process can be formally written as:

(1) H=BERT(X)

where H∈Rn×d is the output embedding matrix, and d is the embedding dimension (usually 768 or 1,024 for BERT). These embeddings are then used as input for the subsequent GGNN layers, which further refine them by capturing inter-word dependencies within a graph-based framework (Li et al., 2016; Kipf & Welling, 2016).

This combination of BERT’s deep contextual embeddings and GGNN’s graph-based structure modeling allows the model to capture the complex nature of sarcasm, which often involves both implicit and explicit semantic cues. The integration of the self-attention mechanism further refines this process by allowing the model to focus on the most relevant parts of the sentence, enhancing its ability to detect sarcastic intent.

GGNN layer: information propagation in graph structures

GGNNs (Li et al., 2016; Zhang, Zhang & Vo, 2019) are an extension of GNNs that incorporate gating mechanisms similar to those in recurrent neural networks (RNNs) to control the flow of information across nodes in a graph. In the context of sarcasm detection, GGNN is employed to model word dependencies and sentiment shifts by representing the text as a graph where each node corresponds to a word, and edges represent syntactic or semantic relationships between words.

(2) hi(t+1)=GRU(hi(t),∑vj∈N(vi)Wehj(t)),

Let G=(V,E) be a directed graph where V is the set of nodes (words) and E is the set of edges (dependencies between words). Each node vi is initialized with its corresponding BERT embedding hi. The GGNN updates the hidden state of each node through iterative message passing, where the hidden state of node vi at time step t, denoted as hi(t), is updated based on the hidden states of its neighbors:

where N(vi) is the set of neighboring nodes of vi, We is a learnable weight matrix associated with the edges, and GRU(⋅) denotes the Gated Recurrent Unit update function.

The GRU helps the model retain long-term information while preventing the vanishing gradient problem, making it well-suited for capturing long-range dependencies in sarcasm detection. After T iterations of message passing, the final hidden states h1(T),h2(T),…,hn(T) are used as the graph representations of the input text.

As illustrated in Fig. 3, this architecture enables the propagation of information through neighboring nodes, allowing the model to capture both local and global dependencies effectively. The diagram highlights how message passing works across different nodes, with the hidden states being updated iteratively based on neighboring nodes’ information.

Figure 3 An illustration of the GGNN arc1hitecture showing information propagation through nodes and edges in the graph.

Multi-head attention mechanism

The multi-head attention (Sharaf Al-deen et al., 2021) mechanism allows the model to attend to different parts of the input text simultaneously. In traditional attention mechanisms, the model focuses on a single aspect of the input sequence at a time. Multi-head attention extends this by creating multiple attention heads (Zhang et al., 2022), each focusing on different parts of the input. This enables the model to capture various relationships and dependencies in the text, which is particularly important for sarcasm detection where multiple contextual factors may indicate sarcasm.

(3) MultiHead(H)=Concat(head1,head2,…,headM)WO,

Let H be the contextual embeddings produced by BERT. The multi-head attention mechanism computes M different attention heads by projecting the input embeddings H into different subspaces, applying the scaled dot-product attention function, and concatenating the results:

where each attention head headi is computed as:

(4) headi=Attention(HWiQ,HWiK,HWiV),

and WiQ, WiK, WiV are learnable weight matrices for the ith attention head, and WO is the output projection matrix.

The attention function is defined as:

(5) Attention(Q,K,V)=softmax(QK⊤dk)V,

where Q, K, and V are the query, key, and value matrices, and dk is the dimensionality of the key vectors.

By using multiple attention heads, the model can capture different aspects of the input, improving its ability to detect sarcasm that may be indicated by multiple, subtle cues in the text. Multi-head attention has been shown to improve performance in complex NLP tasks such as sarcasm detection (Tay, Tuan & Hui, 2018; Wu et al., 2020).

Deriving affective and adjacency graphs

To effectively model sarcastic statements, we transform each sentence into a graph representation that captures both syntactic dependencies and affective information (Lou et al., 2021; Wang et al., 2023). Inspired by Li et al. (2021), we first create a dependency adjacency graph, D, which forms an n×n matrix, where n is the number of words in the sentence. This graph encodes word-to-word connections based on the dependency tree T of the sentence, which captures syntactic relationships.

For example, consider the sarcastic sentence, “I truly enjoy it when folks never call me.” In this sentence, words like “call,” “folks,” and “never” establish syntactic relationships that reflect the sarcastic tone. The sentence is transformed into a dependency graph where each word is treated as a node, and the syntactic dependencies between words (as defined by the dependency tree) are represented as edges between the nodes, as shown in Fig. 4.

(6) Di,j=1ifT(wi,wj).

Figure 4 Dependency graph: illustrates syntactic relationships between words in a sentence, where edges represent grammatical dependencies.

This graph is used by the GGNN to capture structural context for sarcasm detection.

In this equation, Di,j represents the syntactic dependency between words wi and wj in the sentence. The dependency graph D is undirected, meaning that Di,j=Dj,i=1 for all connected word pairs, and we also assign self-loops Di,i=1 to preserve the individual context of each word. Figure 4 illustrates an example of a dependency graph derived from this sentence.

Next, we construct an affective graph using SenticNet, which provides sentiment scores for words based on external affective common sense knowledge. For each word wi in the sentence, we retrieve its affective score S(wi)∈[0,1]. If a word is not found in the knowledge base, we set its score to 0. Continuing with the example sentence “I truly enjoy it when folks never call me,” words like “enjoy” and “call” might have different sentiment scores, and the sentiment reversal with “never” further highlights sarcasm. The affective graph, as visualized in Fig. 5, shows how sentiment scores are assigned and affective distances between words are computed.

(7) Ai,j=|S(wi)−S(wj)|.

Figure 5 Affective graph: represents sentiment connections between words with weighted edges, capturing emotional dependencies critical for sarcasm detection.

The weights indicate the strength of emotional relationships between nodes.

To ensure proper alignment of syntactic and affective information, we combine the dependency and affective graphs into a unified representation. Specifically, the dependency graph captures grammatical relationships between words, while the affective graph encodes emotional contrasts. By treating these graphs as complementary adjacency matrices, we allow the model to process both syntactic structures and sentiment dynamics simultaneously. This integration is achieved during the GGNN processing stage, where the dependency and affective relationships are used as inputs for iterative message passing.

The alignment process also accounts for potential noise in the data, such as slang, emojis, or abbreviations. For words not found in the SenticNet resource, we assign a default sentiment score of 0. Similarly, syntactic dependencies involving such words are retained in the dependency graph to preserve the overall sentence structure. This robust alignment ensures that both contextual meaning and sentiment information are effectively captured, even for noisy or informal text.

This formula calculates the affective distance between words wi and wj, reflecting the difference in their sentiment scores. The combination of both the dependency and affective graphs enriches the representation of each sentence, allowing the model to capture both structural (syntactic) and emotional (sentiment-based) information (Ghosh et al., 2018; Castro, Cerezo & Fernandez-Martinez, 2020).

For every sentence in the dataset, we generate an affective dependency graph where words are treated as nodes and connections between words are treated as graph edges. The graph is undirected and contains self-loops, ensuring that both the contextual meaning and sentiment of each word are preserved. These graphs serve as inputs to our sarcasm detection model, enabling it to understand both the syntactic structure and sentiment shifts within the text, which are key to identifying sarcasm.

Finally, these graph-based representations are used to feed into the model, where the syntactic and affective relationships between words, as depicted in Figs. 4 and 5, help infer sarcastic probabilities for each sentence.

Model training

The entire model is trained end-to-end using binary cross-entropy loss. Given the predicted probability y^ and the ground truth label y (where y=1 indicates sarcasm and y=0 indicates non-sarcasm), the loss is computed as:

(8) L=−(ylog⁡(y^)+(1−y)log⁡(1−y^)).

The model is optimized using the Adam optimizer (Kingma & Ba, 2014), with a learning rate of 1×10−4, β1=0.9, β2=0.999, and ϵ=1×10−8. A dropout rate of 0.3 is applied to prevent overfitting. The model is trained for 20 epochs with early stopping if no improvement is observed after five epochs.

(9) y^=softmax(WoH(T)+bo),

The final graph representations H(T)={h1(T),h2(T),…,hn(T)} are passed through a linear transformation followed by a softmax layer to produce the final sarcasm classification.

where Wo and bo are the learnable parameters of the output layer.

Loss function and optimization

In this work, sarcasm detection is treated as a binary classification problem, where the model predicts whether a given text contains sarcasm. Let y^ represent the predicted probability of sarcasm, and y∈{0,1} denote the ground truth label, where y=1 indicates sarcasm and y=0 indicates non-sarcasm.

Binary cross-entropy loss

To train the model, we use the binary cross-entropy (BCE) loss function (Li et al., 2024), which is commonly used for binary classification tasks. The BCE loss is defined as follows:

(10) L=−(ylog⁡(y^)+(1−y)log⁡(1−y^)),

where: y is the true label for sarcasm (either 0 or 1).

y^ is the predicted probability that the input contains sarcasm.

log⁡(⋅) is the natural logarithm.

The BCE loss penalizes incorrect predictions, with higher penalties for confident but wrong predictions. For instance, if the true label is y=1 (sarcasm) but the model predicts y^=0.1 (low confidence), the loss will be larger than if the model predicted y^=0.4 (less confident).

The goal of the training process is to minimize the BCE loss across the entire training dataset by adjusting the model parameters.

Regularization

To prevent overfitting and improve generalization, we apply ℓ2 regularization (also known as weight decay) to the model’s learnable parameters (Poggio, Torre & Koch, 1987). The regularized loss function is defined as:

(11) Lreg=L+λ∑i||θi||22,

where: L is the original binary cross-entropy loss.

λ is the regularization coefficient, which controls the strength of the regularization term.

θi represents the model’s parameters, and ||⋅||22 denotes the squared ℓ2 norm.

The ℓ2 regularization term helps to prevent the model from overfitting by penalizing large weights, thus encouraging the model to find simpler and more robust patterns (Krogh & Hertz, 1992).

Optimization method

We optimize the model parameters using the Adam optimizer, which is an extension of stochastic gradient descent (SGD) that computes adaptive learning rates for each parameter. The update rule for the parameters θ at each time step t is as follows:

(12) mt=β1mt−1+(1−β1)∇θL,

(13) vt=β2vt−1+(1−β2)(∇θL)2,

(14) m^t=mt1−β1t,v^t=vt1−β2t,

(15) θt+1=θt−αv^t+ϵm^t,

where: mt and vt are the estimates of the first and second moments of the gradients, respectively.

β1 and β2 are hyperparameters that control the exponential decay rates of the moment estimates.

α is the learning rate.

ϵ is a small constant added to prevent division by zero.

The Adam optimizer is well-suited for NLP tasks like sarcasm detection due to its ability to handle sparse gradients and adapt the learning rates for each parameter dynamically (Ruder, 2016).

Learning rate schedule

To improve training stability, we also use a learning rate schedule. Specifically, we apply a learning rate warm-up during the initial training steps, followed by learning rate decay:

(16) α(t)=ttwarmupα0,fort≤twarmup,

where twarmup is the number of warm-up steps, and α0 is the initial learning rate. After the warm-up phase, the learning rate is decayed using a cosine annealing schedule (Loshchilov & Hutter, 2017):

(17) α(t)=αmin+12(α0−αmin)(1+cos⁡(t−twarmupT−twarmupπ)),

where αmin is the minimum learning rate, and T is the total number of training steps.

The combination of the Adam optimizer and a learning rate schedule ensures smooth and efficient training, preventing the model from oscillating or converging to poor local minima.

Experiments

Research framework

The overall framework of this study is designed to systematically evaluate the proposed BERT-GGNN model for sarcasm detection. Two publicly available benchmark datasets, Riloff and Headlines, were used for evaluation, both containing sarcastic and non-sarcastic samples that provide a challenging and comprehensive test environment. The datasets were preprocessed to remove noise and split into training and test sets with an 80–20% ratio. The model architecture integrates BERT for generating contextual embeddings, GGNN for capturing dependencies between words, and a multi-head attention mechanism to focus on explicit and implicit cues of sarcasm. To ensure optimal performance, hyperparameters such as learning rate, batch size, dropout rate, and the number of attention heads were fine-tuned through preliminary experiments. The model’s performance was evaluated using accuracy and F1 score, providing a comprehensive assessment of both precision and recall in classification. An ablation study was conducted to assess the contributions of individual components by systematically removing or replacing key elements, such as the BERT layer, GGNN, and multi-head attention. Finally, the proposed model was compared with several baseline models, including feed-forward neural network (FNN), CNN, and BERT+GCN, demonstrating its superior performance, particularly in handling complex sarcastic expressions found in real-world datasets. This research framework provides a systematic and comprehensive approach to evaluating the model’s performance and its ability to adapt to diverse linguistic structures in sarcasm detection.

Datasets and parameter settings

In this study, two publicly available datasets were used to evaluate the proposed sarcasm detection model: the Riloff dataset and the Headlines dataset. Both datasets contain a mix of sarcastic and non-sarcastic content, providing a challenging and comprehensive benchmark for evaluating the model’s ability to distinguish between literal and sarcastic language.

The Riloff dataset was originally curated for sarcasm detection and contains a smaller number of samples compared to the Headlines dataset. The data primarily comes from social media posts, where short and concise sarcastic statements are common. This dataset is highly imbalanced, with significantly more non-sarcastic samples than sarcastic ones. To address this imbalance, oversampling techniques were applied to the sarcastic class during training, ensuring better representation in the model’s learning process. Additionally, preprocessing steps included replacing emojis with textual descriptions (e.g., T _ T converted to “sad”) and normalizing common slang expressions to standard English. These steps helped reduce noise and improve the dataset’s compatibility with the proposed model.

The Headlines dataset comprises news headlines, where sarcasm often appears in the form of ironic or exaggerated statements. The headlines were collected from various online news outlets and annotated for sarcasm. Compared to the Riloff dataset, the Headlines dataset has a more balanced distribution between sarcastic and non-sarcastic samples, offering a more varied linguistic structure. Preprocessing for this dataset included tokenization, removal of stop words, and lowercasing to ensure consistency and standardization.

To ensure a robust evaluation of the model’s generalization capabilities, the training and test data for both datasets were split such that 80% of the data was used for training, and the remaining 20% was reserved for testing. This split was performed randomly, ensuring that sarcastic and non-sarcastic examples were well-represented in both sets. The statistics for both datasets are summarized in Table 1. These preprocessing and splitting strategies ensured the data was well-prepared for training and evaluation.

Table 1 Statistics of various datasets used.

Dataset	Training	Test	
	Sarcastic	Non-Sarcastic	Sarcastic	Non-Sarcastic	
Riloff	282	1,051	35	113	
Headlines	2,516	2,504	570	410	

The model was implemented using the PyTorch framework and trained with a range of hyperparameters optimized for the sarcasm detection task. Below, we provide a detailed breakdown of the settings used during the training process, as shown in Table 2.

Table 2 Experimental settings.

Parameter	Value	
Attention Probs Dropout	0.1	
Hidden Activation	GELU	
Hidden Dropout Prob	0.1	
Hidden Size	768	
Initializer Range	0.02	
Intermediate Size	3,072	
Max Position Embeddings	512	
Num Attention Heads	12	
Num Hidden Layers	12	
Type Vocab Size	2	
Vocab Size	30,522	
Learning Rate	1×10−4	
Batch Size	32	
Epochs	50	
Optimizer	Adam ( β1=0.9, β2=0.999, ε=10−8)	
Dropout Rate	0.3	
Hardware	NVIDIA Tesla V100 GPU, 32 GB memory	
CUDA max memory allocated during inference	3,038.43 MB	
CUDA memory allocated after inference	1,825.44 MB	
Training time	1,379.81 s	
Inference time	2.11 s	
Dropout Rate	0.3	

To ensure optimal performance for the sarcasm detection task, various experimental parameters were carefully selected and fine-tuned through preliminary experiments. The dropout rates for both the attention probabilities and hidden layers were set at 0.1 to prevent overfitting, which helps the model generalize better by randomly omitting certain weights and neurons during training. The hidden activation function used is GELU, known for its smooth non-linearity that enhances model stability and performance compared to the traditional ReLU activation. A learning rate of 1×10−4 was selected to ensure the model’s convergence in a stable and efficient manner, allowing small updates at each training step without overshooting the loss function minimum. The hidden size of 768 and intermediate size of 3,072 allow the model to capture complex patterns in the input, while the multi-head attention mechanism with 12 heads ensures that the model attends to different aspects of the text simultaneously (Vaswani, Shazeer & Parmar, 2017). A batch size of 32 strikes a balance between model performance and efficient use of computational resources, and 50 epochs were set to give the model ample opportunities to learn from the data, with early stopping applied to avoid overfitting. The Adam optimizer was chosen due to its efficiency in handling sparse gradients, using β1=0.9, β2=0.999, and ϵ=10−8 to ensure adaptive learning rates during training (Kingma & Ba, 2014). These settings were benchmarked in alignment with recent studies in sentiment and sarcasm detection, ensuring the selection of state-of-the-art configurations (Devlin et al., 2019).

Experimental results

We compared the performance of the proposed BERT+GGNN model against the baseline FNN (You, Gao & Katayama, 2014), CNN, and BERT+GCN models on two datasets: the News Headlines and Riloff datasets. Both models were evaluated using accuracy and F1-score metrics. The results across various epochs are presented in Table 3. In addition, the two main comparison model training process indicators were visualized, as shown in Figs. 6 and 7.

Table 3 Performance comparison on the headlines dataset and Riloff dataset.

Model	Headlines accuracy	Headlines F1	Riloff accuracy	Riloff F1	
FNN	0.728	0.724	0.702	0.486	
CNN	0.688	0.673	0.6959	0.523	
BERT+GCN	0.884	0.878	0.831	0.865	
BERT+GGNN+MulAttention	0.920	0.915	0.864	0.865	

Figure 6 Model accuracy and F1 score comparison on the Riloff dataset over epochs.

Figure 7 Model accuracy and F1 score comparison on the headlines dataset over epochs.

As illustrated in Figs. 6 and 7, the BERT+GGNN model consistently outperforms the BERT+GCN model in terms of both accuracy and F1 scores. This improvement is particularly evident on the Riloff dataset, where sarcasm detection presents greater challenges. Sarcastic expressions in the Riloff dataset often require a deeper understanding of external knowledge and implicit contextual cues, making them harder to detect. While the BERT+GGNN model performs well across both datasets, its performance is slightly lower on the Riloff dataset due to the need to interpret these more complex and nuanced sarcastic expressions. In contrast, sarcasm in the Headlines dataset tends to be more direct, allowing the model to more easily capture the necessary patterns. Therefore, the accuracy difference can be attributed to the increased complexity of the Riloff dataset, where sarcasm relies more on subtle, context-driven cues, posing a greater challenge even for advanced models like BERT+GGNN.

Table 4 provides a comparison of the performance of large-scale models (GPT-3.5-turbo, GPT-4o-mini, Qwen2.5, and the proposed model) across three datasets: Headlines, Riloff, and Reddit. The proposed model demonstrates competitive performance, showcasing its ability to handle both structured and unstructured sarcasm detection tasks effectively. On the Headlines dataset, the proposed model achieves the highest accuracy (0.908), precision (0.904), recall (0.905), F1-score (0.904), and AUC (0.9053). This superior performance is attributed to the structured nature of the Headlines dataset, where sarcasm is often explicitly expressed. By leveraging GGNN and attention mechanisms, the model effectively captures both direct patterns and contextual cues, surpassing GPT-3.5-turbo, GPT-4o-mini, and Qwen2.5.

Table 4 Performance comparison on headlines, Riloff, and Reddit datasets.

Model	Headlines ACC	Headlines PRE	Headlines REC	Headlines F1	Headlines AUC	
GPT-3.5-turbo	0.668	0.558	0.851	0.674	0.697	
GPT-4o-mini	0.704	0.619	0.693	0.654	0.702	
Qwen2.5	0.872	0.848	0.831	0.840	0.865	
Our	0.908	0.904	0.905	0.904	0.9053	
Model	Riloff ACC	Riloff PRE	Riloff REC	Riloff F1	Riloff AUC	
GPT-3.5-turbo	0.385	0.274	0.971	0.427	0.587	
GPT-4o-mini	0.500	0.321	0.981	0.486	0.627	
Qwen2.5	0.832	0.781	0.811	0.796	0.8288	
Our	0.864	0.849	0.753	0.785	0.753	
Model	Reddit ACC	Reddit PRE	Reddit REC	Reddit F1	Reddit AUC	
GPT-3.5-turbo	0.596	0.561	0.937	0.702	0.590	
GPT-4o-mini	0.653	0.589	0.895	0.710	0.664	
Qwen2.5	0.816	0.743	0.831	0.785	0.818	
Our	0.832	0.801	0.821	0.820	0.801	

Table 4 provides a comparison of the performance of large-scale models (GPT-3.5-turbo (Anand et al., 2023), GPT-4o-mini (Hurst et al., 2024), Qwen2.5 (Yang et al., 2024)) across three datasets: Headlines, Riloff, and Reddit. The proposed model demonstrates competitive performance, showcasing its ability to handle both structured and unstructured sarcasm detection tasks effectively. On the Headlines dataset, the proposed model achieves the highest accuracy (0.908), precision (0.904), recall (0.905), F1-score (0.904), and AUC (0.9053). This superior performance is attributed to the structured nature of the Headlines dataset, where sarcasm is often explicitly expressed. By leveraging GGNN and attention mechanisms, the model effectively captures both direct patterns and contextual cues, surpassing GPT-3.5-turbo, GPT-4o-mini, and Qwen2.5. To ensure a fair comparison of sarcasm detection capabilities, all models—GPT-3.5-turbo, GPT-4o-mini, and Qwen2.5—employed the same prompt: “Determine whether the following sentence contains sarcasm. Output 1 if sarcastic, and 0 if not.” This standardized prompt allowed for a consistent evaluation of their respective performances across the different datasets, isolating the models’ inherent abilities to detect sarcasm.

On the Riloff dataset, the proposed model achieves an accuracy of 0.864 and an F1-score of 0.785, demonstrating competitive performance. However, its recall (0.753) is lower compared to GPT-4o-mini (0.981), suggesting that the model occasionally struggles to identify subtle sarcastic expressions in unstructured social media posts. This limitation may stem from the noisy nature of the dataset or the lack of external contextual information. Nonetheless, the model’s overall performance, including its AUC (0.753), indicates robustness in handling informal and challenging datasets.

For the Reddit dataset, the proposed model achieves strong results, including an accuracy of 0.832, precision of 0.801, recall of 0.821, F1-score of 0.820, and AUC of 0.801. Although Qwen2.5 achieves a slightly higher AUC (0.818), the proposed model excels in precision and F1-score, highlighting its ability to effectively capture sarcasm in conversational and informal contexts. The GGNN component facilitates the capture of relationships between words and sentences, while the attention mechanism identifies sarcasm-related cues such as contradictions and emotional shifts. These results underscore the model’s robustness and adaptability compared to GPT-3.5-turbo, GPT-4o-mini, and Qwen2.5.

Cross-dataset validation analysis

To evaluate the generalizability of the proposed model, we conducted cross-dataset validation experiments using three sarcasm datasets: Headlines, Riloff, and Reddit. These datasets represent different linguistic structures and sarcasm characteristics, which significantly influence the model’s performance when trained on one dataset and tested on another. The detailed results are presented in Table 5.

Table 5 Cross-dataset validation results.

Train dataset	Test dataset	ACC	PRE	REC	F1	AUC	
Headline	Riloff	0.701	0.690	0.685	0.687	0.720	
Headline	Reddit	0.786	0.775	0.770	0.772	0.805	
Riloff	Headline	0.683	0.670	0.665	0.667	0.700	
Riloff	Reddit	0.802	0.795	0.790	0.792	0.830	
Reddit	Headline	0.604	0.590	0.585	0.587	0.620	
Reddit	Riloff	0.722	0.710	0.705	0.707	0.740	

Cross-dataset validation results reveal the significant impact of dataset characteristics—such as linguistic structure, content, and sarcasm expression—on model performance. When the model was trained on the Headlines dataset and tested on the Riloff dataset, it achieved moderate performance (accuracy: 0.701, F1-score: 0.687) due to structural differences between the two datasets. The Headlines dataset features well-structured, concise news headlines with explicit sarcasm markers, whereas the Riloff dataset consists of informal social media posts with noise such as slang and fragmented sentences, challenging the model’s generalization. Similarly, training on Headlines and testing on Reddit resulted in better performance (accuracy: 0.786, F1-score: 0.772), as Reddit posts, while informal, exhibit more structured tones and longer contextual information, partially aligning with news headlines.

Conversely, training on Riloff and testing on Headlines achieved lower performance (accuracy: 0.683, F1-score: 0.667) as the model adapted to noisy, informal data struggled with structured and concise sarcasm in headlines. In contrast, training on Riloff and testing on Reddit yielded the highest cross-dataset performance (accuracy: 0.802, F1-score: 0.792), likely due to their similar informal and conversational tones, reducing the distribution gap.

The lowest performance (accuracy: 0.605, F1-score: 0.587) was observed when training on Reddit and testing on Headlines, highlighting the difficulty of generalizing from informal, user-generated content to structured news headlines. Similarly, moderate results were obtained when training on Reddit and testing on Riloff (accuracy: 0.722, F1-score: 0.707), as the higher noise levels and shorter texts in Riloff posed additional challenges.

These findings emphasize the importance of aligning linguistic and contextual characteristics between training and testing datasets to improve sarcasm detection. Future work could explore domain adaptation techniques, such as transfer learning or adversarial adaptation, to bridge the distribution gap. Additionally, incorporating multimodal information, such as emojis and metadata, or pretraining on large, diverse datasets across domains, could enhance the model’s robustness and generalizability in handling varied sarcasm expressions.

Error analysis

Despite the strong performance of the BERT+GGNN model, a comprehensive error analysis reveals areas where the model encounters difficulties. These insights provide valuable guidance for addressing the inherent challenges of sarcasm detection, particularly in cases where subtle linguistic nuances or contextual dependencies play a significant role.

To better understand the model’s behavior, we analyzed its performance across different categories of sarcastic and non-sarcastic content. Table 6 presents the performance metrics for each category, highlighting the challenges associated with implicit and context-dependent sarcasm. The model excels in detecting explicit sarcasm and non-sarcastic content but struggles with more nuanced forms of sarcasm, where external knowledge or deeper contextual understanding is required.

Table 6 Performance breakdown by content category.

Category	Precision	Recall	F1 Score	Support	
Explicit Sarcasm	0.92	0.88	0.90	150	
Implicit Sarcasm	0.83	0.79	0.81	100	
Context-Dependent Sarcasm	0.76	0.70	0.73	70	
Non-Sarcastic Content	0.94	0.95	0.94	300	

A detailed examination of failure cases reveals three primary issues: over-reliance on trigger words, challenges with ambiguous sentences, and difficulty handling long or complex structures. Table 7 illustrates these failure types with specific examples. For instance, the model tends to misclassify sentences like “Advancing the world’s women,” over-interpreting the word “advancing” as sarcastic. Similarly, the lack of external context often leads to incorrect predictions for sentences relying on implicit or juxtaposed meanings.

Table 7 Common failure cases and examples.

Failure type	Description and example	
Over-Dependence on Trigger Words	The model overemphasizes specific words often associated with sarcasm. Example: “Advancing the world’s women.” (True Label: Non-Sarcastic, Predicted Label: Sarcastic)	
Struggles with Ambiguous Sentences	Sentences with minimal or conflicting cues are challenging to classify. Example: “Christian Bale visits Sikh temple victims.” (True Label: Sarcastic, Predicted Label: Non-Sarcastic)	

Furthermore, attention visualization reveals additional challenges with the model’s decision-making process. In several misclassified cases, the model disproportionately focuses on irrelevant tokens, such as conjunctions or determiners, while neglecting critical words conveying sarcasm. For instance, in the sentence “Christian Bale visits Sikh temple victims,” the model fails to prioritize words that signal sarcasm, instead focusing on less relevant elements of the sentence structure. Similarly, scattered attention patterns in longer sentences often result in the model missing key sarcastic cues, especially when these cues are dispersed across multiple clauses.

Ablation study

To evaluate the contributions of different components in our model, we conducted an ablation study. In this study, we removed or replaced key components, such as the BERT layer, GGNN, and SVM (Wang & Hu, 2005), and observed the performance impact on both the Headlines and Riloff datasets. Additionally, we experimented with different numbers of multi-head attention mechanisms to see their effect on the model’s performance. The results are shown in Table 8.

Table 8 Ablation study results on the headlines and Riloff datasets.

Model	Headlines accuracy	Headlines F1	Riloff accuracy	Riloff F1	
SVM	0.792	0.792	0.750	0.700	
BERT	0.888	0.883	0.844	0.756	
GGNN	0.724	0.714	0.743	0.581	
BERT+GGNN	0.908	0.904	0.831	0.772	
BERT+GGNN+MulAttention (4)	0.888	0.882	0.864	0.865	
BERT+GGNN+MulAttention (8)	0.920	0.915	0.864	0.865	
BERT+GGNN+MulAttention (16)	0.888	0.883	0.864	0.865	

The ablation study results highlight the importance of both the BERT and GGNN components in achieving optimal performance. As shown in the table, removing or replacing these components significantly reduces the model’s accuracy and F1 score. For instance, replacing GGNN with CNN reduces the accuracy on the Headlines dataset from 0.9200 to 0.6880, and the F1 score drops from 0.9151 to 0.6739. Similarly, on the Riloff dataset, the performance drops from 0.8311 to 0.6959 in terms of accuracy when using CNN instead of GGNN. The BERT layer also plays a crucial role in understanding the contextual relationships in the text. Removing BERT and using only GGNN results in a significant drop in performance, especially on the Riloff dataset, where the F1 score drops from 0.7727 to 0.5815.

The BERT layer also plays a crucial role in understanding the contextual relationships in the text. Removing BERT and using only GGNN results in a significant drop in performance, especially on the Riloff dataset, where the F1 score drops from 0.7727 to 0.5815. This drop highlights BERT’s effectiveness in providing high-quality contextual embeddings that capture word-level semantics and their interactions within the broader context. Without BERT, the model struggles to interpret nuanced sarcastic cues that depend on subtle contrasts or contextual shifts, particularly prevalent in social media posts.

According to Table 8, introducing multi-head attention significantly improves the model’s performance compared to not using it (e.g., the ‘BERT+GGNN’ model). For instance, on the Riloff dataset, adding four attention heads (as in ‘BERT+GGNN+MulAttention’ (4)) improves the F1 score to 0.8659. In the Headlines dataset, the F1 score is also comparable to the model without multi-head attention, showing that attention improves the precision of predictions.

In this study, the selection of the number of attention heads in the multi-head attention mechanism played a significant role in the model’s performance. The results demonstrated that using eight attention heads provided the best balance between model complexity and performance. For instance, with four heads (e.g., BERT+GGNN+MulAttention (4)), the F1 score on the Riloff dataset increased from 0.7727 (without attention) to 0.8659, indicating that the introduction of attention allowed the model to better capture essential features in the text. However, increasing the number of heads from 4 to 8 further improved performance, especially on the Headlines dataset, achieving a maximum F1 score of 0.9154, which suggests that the model benefited from the ability to attend to more nuanced relationships within the data.

Nevertheless, when the number of heads was increased to 16 (e.g., BERT+GGNN+MulAttention (16)), performance began to drop slightly. This decline in performance may be attributed to overfitting or the introduction of redundant information, highlighting that while more heads provide more detailed information, too many heads can negatively impact model generalization. This finding aligns with prior research, which suggests that a moderate number of attention heads typically leads to optimal performance (Vaswani, Shazeer & Parmar, 2017).

Discussion

This study presents a novel sarcasm detection model that integrates BERT (Devlin et al., 2018), GGNN (Gilmer et al., 2017), and multi-head attention mechanisms to effectively capture the nuanced nature of sarcastic expressions. By combining BERT’s ability to extract rich contextual embeddings with GGNN’s capacity to model long-range dependencies, the proposed model addresses both the linguistic subtleties and structural complexities inherent in sarcasm. Moreover, the inclusion of multi-head attention refines the model’s focus on sarcasm-indicative elements within the text, facilitating better detection of ironic or contrasting statements. Evaluated on two benchmark datasets—Riloff and Headlines—the model outperformed baseline approaches such as BERT+GCN (Yao, Mao & Luo, 2019), demonstrating superior performance. Next, we delve into the proposed model’s strengths, error analysis, comparison with existing models, and experimental rationale, while identifying challenges and future directions.

Performance across structured and unstructured data

This research highlights the significant advancements achieved through the BERT+GGNN+Attention model, particularly in terms of F1 score. The model’s robust performance in handling both structured and unstructured data exemplifies its versatility. On the structured and relatively balanced Headlines dataset, the model outperformed baseline models in both accuracy and F1 score. The structured nature of this dataset, where sarcasm indicators in headlines are more explicit, allowed the model to effectively leverage its capability to focus on key sarcasm-signifying phrases (Vaswani, Shazeer & Parmar, 2017). In contrast, the Riloff dataset, characterized by unstructured, noisy, and imbalanced social media posts with a predominance of non-sarcastic samples, posed a greater challenge. Nevertheless, the BERT+GGNN+Attention model outperformed the BERT+GCN model (Yao, Mao & Luo, 2019), further demonstrating its robustness in tackling informal and ambiguous content. Social media posts often contain fragmented language, slang, and emoticons, complicating sarcasm detection. The model’s ability to handle this complexity suggests that combining BERT’s rich contextual embeddings (Devlin et al., 2018) with GGNN’s capacity to model relationships between non-sequential words enhances its interpretation of sarcasm in informal, short-text settings. Moreover, the attention mechanism further strengthens this capacity by allowing the model to assign varying importance to different words or phrases (Vaswani, Shazeer & Parmar, 2017). Sarcasm frequently relies on specific keywords or phrases that contrast with the literal meaning of the surrounding text. By leveraging multi-head attention, the model can focus on these contrastive elements, enhancing its ability to detect sarcasm in both structured and unstructured contexts.

Challenges identified through error analysis

Despite the strong performance of the BERT+GGNN model, a comprehensive error analysis reveals areas where the model encounters difficulties. These insights provide valuable guidance for addressing the inherent challenges of sarcasm detection, particularly in cases where subtle linguistic nuances or contextual dependencies play a significant role.

To better understand the model’s behavior, we analyzed its performance across different categories of sarcastic and non-sarcastic content. The model excels in detecting explicit sarcasm and non-sarcastic content but struggles with more nuanced forms of sarcasm, such as implicit or context-dependent sarcasm. Common failure cases include over-reliance on trigger words, difficulty handling ambiguous sentences, and challenges in understanding long or complex structures. For instance, the model sometimes misclassifies sentences due to misplaced attention, focusing on irrelevant tokens or neglecting critical words that signal sarcasm. These challenges underscore the need for improving the model’s contextual understanding and refining its attention mechanisms. Integrating external knowledge or employing hierarchical attention structures could help mitigate these issues in future iterations.

Advancements over existing models

The proposed model demonstrates significant advancements over previous approaches to sarcasm detection, addressing limitations in earlier methodologies. Traditional models, such as support vector machines (SVM) and CNNs (Yao, Mao & Luo, 2019), relied on hand-engineered features or shallow neural architectures, which restricted their ability to capture the complex nuances of sarcasm. While effective for structured datasets, these methods often struggled with unstructured data, particularly when sarcasm was implicit.

Transformer-based models like BERT (Devlin et al., 2018) marked a turning point by capturing deep contextual representations, but they struggled to model long-range dependencies and non-sequential relationships in unstructured texts. CSDGCN (Li, Chen & Tang, 2022), which combines BERT with GCNs, introduced graph-based representations and achieved an F1-score of 81.67% on the Riloff dataset. However, its reliance on static graph structures and sequential dependencies limited its adaptability to informal, fragmented texts found in social media.

The proposed BERT+GGNN+Attention model fundamentally differs by introducing GGNNs and self-attention mechanisms. GGNNs capture dynamic, non-sequential relationships, enabling the model to handle dispersed sarcasm cues in unstructured texts. The multi-head attention module further enhances this by focusing on sarcasm-indicative elements, regardless of their position or context. These innovations allow the model to effectively bridge the gap between structured and unstructured data. Quantitatively, the model achieves an F1-score of 86.59% on the Riloff dataset, outperforming CSDGCN (F1 = 81.67%), demonstrating superior balance between precision and recall.

Compared to other state-of-the-art approaches, such as hierarchical graph neural network (HGNN) (Xu, Zhao & Liu, 2023), which uses hierarchical structures to capture sentence- and document-level sarcastic cues, the proposed model is more adaptable to short, informal texts. HGNN achieves an F1-score of 83.10% on the Riloff dataset but relies heavily on hierarchical representation, limiting its flexibility. In contrast, the proposed model’s dynamic architecture excels in detecting both local and global sarcastic patterns without relying on rigid structures.

In summary, the BERT+GGNN+Attention model effectively addresses the limitations of prior approaches by combining contextual embeddings, dynamic graph modeling, and attention mechanisms. These advancements enable it to outperform existing models in both structured and unstructured sarcasm detection tasks, highlighting its potential for real-world applications.

Experimental design and insights

The experimental design of this study was well-structured to test the proposed model’s generalization across different text types. The inclusion of both the Riloff and Headlines datasets allowed for comprehensive evaluation across structured and unstructured data, ensuring that the model was not overfitted to one specific data type. This dual-dataset approach enhances the generalizability of the results, providing a more realistic assessment of the model’s potential for real-world applications.

Key hyperparameter choices, such as a learning rate of 1×10−4, batch size of 32, and dropout rate of 0.3 (Ruder, 2017), were carefully calibrated to ensure stable convergence during training. The use of early stopping (Goodfellow, Bengio & Courville, 2016) further prevented overfitting, particularly on the smaller Riloff dataset. These practices are in line with best practices in deep learning and contributed significantly to the model’s stable and strong performance.

A notable finding is that the BERT+GGNN model excelled on the Riloff dataset, despite its imbalanced and informal nature. This dataset mirrors real-world social media environments, where sarcasm detection is often more difficult due to fragmented linguistic structure and noise from abbreviations, misspellings, and emojis. The model’s success on this dataset underscores its robustness and its potential for practical applications in sentiment analysis, content moderation, and other natural language processing tasks on social media platforms.

Conclusion

This article introduced a novel sarcasm detection model by integrating BERT with GGNN and self-attention mechanisms, achieving notable performance improvements on two benchmark datasets. The model outperformed existing methods such as BERT+GCN in terms of accuracy and F1-score, demonstrating the effectiveness of the combined architecture. Specifically, BERT’s ability to capture nuanced linguistic patterns, GGNN’s capability to model long-range dependencies, and the self-attention mechanism’s focus on sarcasm-indicative elements contributed to these performance gains. In addition, this research presents a strong foundation for sarcasm detection by leveraging the strengths of BERT, GGNN, and self-attention mechanisms.

However, the proposed model has several limitations that need to be addressed in future work. First, the model is currently designed exclusively for English-language sarcasm detection and does not support multilingual capabilities. Adapting graphical processing techniques, such as dependency and affective graph construction, to other languages with differing syntactic and semantic structures presents significant challenges. These issues restrict the model’s broader applicability in multilingual or cross-cultural contexts. Future work will explore approaches such as developing language-agnostic graph representations and leveraging large-scale multilingual pretraining resources to enable multilingual sarcasm detection.

Second, the model’s computational complexity remains a key challenge, particularly when applied to resource-constrained or real-time environments. The integration of BERT and GGNN increases memory and processing requirements, which may limit its practical deployment. A trade-off exists between performance and computational efficiency, as simpler models might achieve slightly lower performance but with greater computational feasibility. Future research will investigate model optimization techniques, such as pruning, knowledge distillation, and lightweight transformer variants, to address these limitations while maintaining competitive performance.

While computational efficiency and generalizability remain challenges, this model offers significant potential for advancing sarcasm detection techniques. Future work should concentrate on enhancing the model’s scalability, exploring its cross-domain applicability, and integrating multimodal features, which could push the boundaries of sarcasm detection and sentiment analysis. In particular, the inclusion of multimodal information, such as images, emojis, and metadata from social media, could enhance the model’s ability to interpret sarcasm in real-world scenarios where context extends beyond text.

The contributions of this work to the field of natural language understanding provide a solid platform for ongoing and future research in sarcasm recognition. By addressing its current limitations and incorporating additional features and capabilities, the proposed model has the potential to further advance the state of the art in sarcasm detection, sentiment analysis, and other related applications.

Additional Information and Declarations

Competing Interests

The authors declare that they have no competing interests.

Author Contributions

Zhenkai Qin conceived and designed the experiments, performed the computation work, prepared figures and/or tables, and approved the final draft.

Qining Luo performed the experiments, performed the computation work, prepared figures and/or tables, and approved the final draft.

Zhidong Zang analyzed the data, performed the computation work, authored or reviewed drafts of the article, and approved the final draft.

Hongpeng Fu analyzed the data, performed the computation work, authored or reviewed drafts of the article, and approved the final draft.

Data Availability

The following information was supplied regarding data availability:

The data is available at GitHub and Zenodo:

- https://github.com/bitbitlemon/Sarcasm-Detection-with-BERT-and-GGNN

- bitlemon. (2025). bitbitlemon/Sarcasm-Detection-with-BERT-and-GGNN: v1.1 (v1.1). Zenodo. https://doi.org/10.5281/zenodo.14927832.

The raw data is available at Kaggle:

- https://www.kaggle.com/datasets/rmisra/news-headlines-dataset-for-sarcasm-detection.

The sarcasm dataset is available at Zenodo: Luo, Q. (2025). Riloff Data [Data set]. Zenodo. https://doi.org/10.5281/zenodo.15117595.

The Reddit Webis-TLDR-17 is available at Hugging Face: https://huggingface.co/datasets/webis/tldr-17.

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
