# Peer review of "Detecting sarcasm in user-generated content integrating transformers and gated graph neural networks"

_PeerJ Computer Science, doi:10.7717/peerj-cs.2817_

## Round 0.1 · original submission · Major Revisions

Dear Authors,
Your paper has been revised. Based on the reviewers' evaluation, major revisions are needed before it is considered for publication in PEERJ Computer Science. More precisely, the following points must be faced in the revised version of your paper:

1) The manuscript lacks a detailed discussion of the model's weaknesses. For instance, while the proposed BERT+GGNN approach achieves competitive performance, an error analysis is absent. Conduct an in-depth error analysis, identifying common failure cases (e.g., over-dependence on specific sarcastic cues, struggles with ambiguous sentences).
2) The manuscript claims superior performance on the Riloff dataset (F1 = 86.59), outperforming CSDGCN (F1 = 81.0). However, no direct comparison with this model is provided. Include a detailed comparison of your approach with CSDGCN and other state-of-the-art GNN-based models. Discuss both quantitative metrics and architectural differences to substantiate your claims. Additionally, provide open access to your code for reproducibility, as is standard in the field.

Reviewer 1 ·

Basic reporting

This paper presents a novel sarcasm detection model that combines Bidirectional Encoder Representations from Transformers (BERT) with Gated Graph Neural Networks (GGNN) and a self-attention mechanism to effectively capture both contextual semantics and global relationships in text. The model employs dependency and affective graphs to enhance its understanding of syntactic and emotional nuances, achieving state-of-the-art results on the Riloff and Headlines datasets with F1 scores of 86.59 and 91.51, respectively. Through ablation studies, the effectiveness of each component—BERT, GGNN, and multi-head attention—is validated. The proposed approach significantly outperforms baseline methods, offering a robust solution for sarcasm detection, though computational efficiency and generalizability to multilingual contexts remain challenges.

However, here are some comments:
(1) The proposed integration of BERT and GGNN is a commendable approach for sarcasm detection, yet combining transformers with GNNs is not novel. Recent works like M3GAT and CSDGCN (2023) have explored similar pathways.
(2) Clearly articulate how the proposed approach differs. For example, does the self-attention mechanism or specific graph construction methodology offer a unique advantage? A detailed comparison with M3GAT and its attention aggregation method would clarify the novelty of your work.

Experimental design

(3) The manuscript largely focuses on conventional sarcasm detection models, overlooking recent advances in large language models (LLMs). While the proposed method is evaluated against BERT, it does not contextualize its relevance in the era of advanced LLMs like GPT-4 or Claude.
(4) Expand the Related Work section to include a thorough discussion on sarcasm detection capabilities in LLMs.

Validity of the findings

(5) The manuscript lacks a detailed discussion of the model's weaknesses. For instance, while the proposed BERT+GGNN approach achieves competitive performance, an error analysis is absent. Conduct an in-depth error analysis, identifying common failure cases (e.g., over-dependence on specific sarcastic cues, struggles with ambiguous sentences).
(6) The manuscript claims superior performance on the Riloff dataset (F1 = 86.59), outperforming CSDGCN (F1 = 81.0). However, no direct comparison with this model is provided. Include a detailed comparison of your approach with CSDGCN and other state-of-the-art GNN-based models. Discuss both quantitative metrics and architectural differences to substantiate your claims. Additionally, provide open access to your code for reproducibility, as is standard in the field.

Reviewer 2 ·

Basic reporting

The paper proposes a novel sarcasm detection
model that combines Bidirectional Encoder Representations from Transformers (BERT) with Gated Graph Neural Networks (GGNN)

1. The paper is written in clear, professional English, making it accessible to an international audience. However, minor grammatical corrections can enhance readability (e.g., inconsistent use of tense and unclear phrases in the introduction).

2.Recent studies (e.g., from 2023) in multimodal sarcasm detection are missing.

3. The introduction provides an excellent overview of sarcasm detection challenges and advances in NLP, situating the research within its field. Additional emphasis on prior GNN and Transformer hybrid studies could enhance this section.

4. Figure 5 (Affective Graph) could include a brief textual explanation.

Experimental design

1. While the BERT and GGNN combination is mentioned, the exact role and interaction between these components could be explained a bit more clearly. For example, how exactly the self-attention mechanism enhances the model could be expanded upon.

2. The integration of BERT for contextual embeddings and GGNN for global semantic structures is innovative and clearly explained. Including a schematic diagram of the complete architecture workflow with intermediary outputs would help readers visualize the end-to-end process.

3. The preprocessing steps for noisy data, such as handling emojis, slang, and abbreviations, are not described in detail. Explicitly stating these steps would clarify how the data aligns with the proposed model.

4. The use of accuracy and F1 score is standard and effective. Including additional metrics like precision, recall, and area under the ROC curve (AUC) would provide a more comprehensive evaluation, particularly for the imbalanced Riloff dataset.

5. Evaluating the model's generalization capability on additional sarcasm detection datasets, especially those from diverse domains (e.g., product reviews, political debates), would bolster the claim of robustness.

6. The paper mentions a training time of approximately two hours per model on an NVIDIA Tesla V100 GPU. Including more details about the computational costs, such as memory usage and time for inference, would provide clarity on the model's feasibility for real-world applications.

Validity of the findings

2. While the paper explains the overall performance metrics, a breakdown of model performance across different categories of sarcastic and non-sarcastic content would add depth. For example:
Which types of sarcasm (e.g., explicit, implicit, context-dependent) are most accurately detected?
Are there specific error patterns that the model frequently encounters?

2. Discussing whether the proposed model architecture can be adapted to multilingual sarcasm detection would help validate its broader applicability.

3. The results are compelling, showing improvements over baseline models. However, including a discussion on why specific improvements occurred (e.g., GGNN handling long-range dependencies or attention mechanisms identifying critical sarcastic cues) would better justify the findings.

4. Testing the model on an independent, unseen sarcasm dataset would validate its generalization ability. For example, applying the trained model to a dataset from a different domain (e.g., customer reviews or political tweets) would strengthen claims of robustness.
5. A discussion on the trade-off between performance and computational efficiency would help validate its practicality in real-world scenarios, where simpler models might suffice for marginally lower performance.

Additional comments

Nil

---

## Round 0.2 · Minor Revisions

Dear Authors,
Your paper has been revised. It needs minor revisions before being accepted for publication. More precisely:

(1) You should citate the M3GAT network.
(2) You should indicate what is the prompt for GPT-3.5, 4 and Qwen 2.5 chatbots.

Reviewer 1 ·

Basic reporting

Only two comments here:
(1) The M3GAT has no citation, dont know why.
(2) What is the prompt for GPT-3.5, 4 and Qwen 2.5? the author should present them in the paper, instead of just listing the results. In addition, these models have also no citation, why?

Experimental design

N/A

Validity of the findings

N/A

Reviewer 2 ·

Basic reporting

The comments are addressed in the revised manuscript.

Experimental design

The comments are addressed in the revised manuscript.

Validity of the findings

The comments are addressed in the revised manuscript.

Additional comments

The comments are incorporated in the revised manuscript. Formatting corrections need to be made. Few tables are going beyond the margin.

---

## Round 0.3 · accepted · Accept

Dear Authors,
Your paper has been accepted for publication in PEERJ Computer Science. Thank you for your fine contribution.